# CC-Chemokine Receptor-2 Expression in Osteoblasts Contributes to Cartilage and Bone Damage during Post-Traumatic Osteoarthritis

**DOI:** 10.3390/biom13060891

**Published:** 2023-05-26

**Authors:** Helen Willcockson, Huseyin Ozkan, José Valdés-Fernández, Liubov Arbeeva, Esra Mucahit, Layla Musawwir, Lola B. Hooper, Froilán Granero-Moltó, Felipe Prósper, Lara Longobardi

**Affiliations:** 1Division of Rheumatology, Allergy and Immunology, University of North Carolina at Chapel Hill, 3300 Thurston Bldg, Campus Box 7280, Chapel Hill, NC 27599, USA; helen_willcockson@med.unc.edu (H.W.); ozkan@email.unc.edu (H.O.); liubov_arbeeva@med.unc.edu (L.A.); esranm@email.unc.edu (E.M.); laylamus@live.unc.edu (L.M.); lolableu@email.unc.edu (L.B.H.); 2Program of Regenerative Medicine, Center for Applied Medical Research (CIMA), Universidad de Navarra, 31008 Pamplona, Spain; jvaldes.3@alumni.unav.es (J.V.-F.); fgranero@unav.es (F.G.-M.); fprosper@unav.es (F.P.); 3Department of Orthopedic Surgery and Traumatology, Clínica Universidad de Navarra, 31008 Pamplona, Spain; 4Instituto de Investigacion Sanitaria de Navarra (IdiSNA), 31008 Pamplona, Spain; 5Department of Hematology and Cell Therapy and CCUN, Clínica Universidad de Navarra, 31008 Pamplona, Spain; 6CIBERONC, 28029 Madrid, Spain; 7Program of Hemato-Oncology, Center for Applied Medical Research (CIMA), Universidad de Navarra, 31008 Pamplona, Spain

**Keywords:** chemokines, bone, osteoarthritis

## Abstract

In osteoarthritis (OA), bone changes are radiological hallmarks and are considered important for disease progression. The C-C chemokine receptor-2 (CCR2) has been shown to play an important role in bone physiology. In this study, we investigated whether *Ccr2* osteoblast-specific inactivation at different times during post-traumatic OA (PTOA) progression improves joint structures, bone parameters, and pain. We used a tamoxifen-inducible *Ccr2* inactivation in Collagen1α-expressing cells to obtain osteoblasts lacking *Ccr2* (*CCR2*-*Col1αKO*). We stimulated PTOA changes in *CCR2*-*Col1αKO* and *CCR2*+/+ mice using the destabilization of the meniscus model (DMM), inducing recombination before or after DMM (early- vs. late-inactivation). Joint damage was evaluated at two, four, eight, and twelve weeks post-DMM using multiple scores: articular-cartilage structure (ACS), Safranin-O, histomorphometry, osteophyte size/maturity, subchondral bone thickness and synovial hyperplasia. Spontaneous and evoked pain were assessed for up to 20 weeks. We found that early osteoblast-*Ccr2* inactivation delayed articular cartilage damage and matrix degeneration compared to *CCR2*+/+, as well as DMM-induced bone thickness. Osteophyte formation and maturation were only minimally affected. Late Collagen1α-*Ccr2* deletion led to less evident improvements. Osteoblast-*Ccr2* deletion also improved static measures of pain, while evoked pain did not change. Our study demonstrates that *Ccr2* expression in osteoblasts contributes to PTOA disease progression and pain by affecting both cartilage and bone tissues.

## 1. Introduction

In osteoarthritis (OA), bone changes such as osteophyte formation and subchondral bone sclerosis are radiological hallmarks and are considered important for disease progression. Increased expression of chemokines in joint tissues is believed to be linked to OA progression; in particular, the C-C chemokine receptor-2 (CCR2) and its ligands CCL2 and CCL12 (a.k.a. MCP1 and MCP5, respectively) that are known to be involved in the recruitment of immune cells at sites of inflammation [1,2], have been shown to be significantly increased in both humans with OA and in rodent models of OA and to mediate OA pain [2,3,4,5,6,7,8,9,10]. In addition to macrophages, CCR2, CCL2, and CCL12 are expressed on chondrocytes, osteoblasts, and tendon fibroblasts, where they influence early tissue degeneration following injury [6]. Thus, the relevance of the CCR2 signaling for potential OA therapies is not limited to mediating the inflammatory response accompanying disease progression [4,5,6] but, differently from other chemokines, is directly linked to cartilage and bone integrity.

Several studies suggest a critical contribution of the CCR2 axis to both osteoblast and osteoclast physiology: increased bone mass was measured in the germline *Ccr2 null* mice and associated with resistance to bone loss in an osteoporotic challenge, thought to be secondary to reduced osteoclastogenesis [11], a mechanism which may also lead to the delayed healing in a fracture model [12]. In contrast, in two well-established murine rheumatoid arthritis (RA) models, *Ccr2 null* mice showed enhanced bone erosion and osteoclast activity leading to more severe disease than wild-type mice, accompanied by marked infiltration of inflammatory cells [13]. A study on orthodontic tooth movement showed that *Ccr2 null* mice, in spite of decreased osteoclast number in the periodontal tissue, exhibited decreased osteoblast differentiation, as determined by decreased mRNA expression of *Collagen 1* and *Osteocalcin* [14]. Although the authors suggest that reduced osteoblast activity may be linked to a decrease in osteoclast stimulatory signals, they acknowledge the need to address a potential direct role of CCR2 in osteoblasts. Indeed, *Ccr2* expression has been detected in human osteoblasts and their precursors by RT-PCR, with levels increasing in pathological conditions [15]. Importantly, microarray data have shown *Ccr2* up-regulation in OA bone tissues [16]. In previous mouse studies, using the destabilization of the medial meniscus (DMM) murine model of post-traumatic OA (PTOA), we have detected increased protein levels of CCR2 and CCL12 in the periosteum and in cortical bone osteoblasts and have shown a critical role for CCR2 in bone sclerosis during injury-induced-OA [6]. In particular, we demonstrated that blockage of the CCR2 signaling during PTOA using a small receptor antagonist (RS504393) had protective effects on both cartilage- and bone-related OA changes, either when the antagonist was delivered systemically at early PTOA stages or by intraarticular deposition using microplates for sustained release [6,9]. Additional DMM studies by our group showed that a tissue-specific deletion of the *Ccr2* gene in aggrecan-expressing cells at the time of injury was beneficial on articular cartilage structure and pain but was less effective on bone damage, suggesting that *Ccr2* expression in the osteoblast niche might contribute to the OA bone changes induced by injury.

The purpose of this study is to determine the role of *Ccr2* expression in osteoblasts during PTOA and analyze both its preventive and therapeutic action to the development of whole joint degeneration and whether *Ccr2* expression in the bone niche contributes to PTOA pain. To achieve this goal, we used an inducible tissue-specific inactivation of *Ccr2* in osteoblasts, inducing recombination before DMM (to analyze its preventive action on disease onset) or after four weeks post-injury (to analyze its action on OA progression), and followed PTOA changes in cartilage and bone tissues at different stages. This study is critical to identify PTOA targets for therapeutic intervention for both joint structure and pain.

## 2. Materials and Methods

### 2.1. Materials and Antibodies

Rabbit polyclonal anti-CCR2 was purchased from Novusbio (NBP2-67700; Centennial, CO, USA; used at 1:100); Goat polyclonal anti-GFP (ab6673; used at 1:100) was purchased from Abcam (Cambridge, MA, USA). Immunohistochemistry was performed using a biotinylated secondary antibody (anti-goat, 705-065-147 or anti-rabbit, 711-065-151; Jackson Immunoresearch, West Grove, PA, USA) and an avidin-biotin complex visualized with diaminobenzidine (Vectastain Elite ABC kit, PK6100; DAB substrate, SK-4100; Burlingame, CA, USA).

### 2.2. Animals

Animal use protocols followed ARRIVE guidelines and were approved by UNC Animal Care and Use Committee. *Collagen1α1*-*CreER* (*Col1α1CreER*) mice have an osteoblast-specific enhancer element. Indeed, transgenic mice harboring a 2.3-kb proximal fragment of col1a1 promoter showed high activity of the transgene in bone-forming cells, osteoblasts, and teeth, very low activity in tendons, and no activity in other tissues [17]. In the *CCR2*-floxed mice (*CCR2^flx^*^/*flx*^*eGFP*), the exon 3 of the *Ccr2* allele is flanked by loxP sites, followed by an eGFP cassette [18]. All animals were on a C57BL/6 background.

*Col1α1*-*CreER* mice were crossed with *CCR2^flx^*^/*flx*^*eGFP* for two generations. The resulting *CCR2^flx^*^/*flx*^-*eGFP*/*Col1α1CreER*-positive and *CCR2^flx^*^/*flx*^-*eGFP*/*Col1α1CreER*-negative were used for PTOA experiments. We injected tamoxifen (40µg/g of weight) in *CCR2^flx^*^/*flx*^-*eGFP*/*Col1α1CreER*-positive to induce Cre recombination and obtained a mouse line in which *Ccr2* is ablated in osteoblasts (referred to as *CCR2*-*Col1α1KO*). Recombination was achieved at two different time points: 2 weeks (wks) before surgery or 4 wks after surgery (to obtain an early or late *Ccr2* ablation, respectively). Animals were euthanized at 2, 4, 8, and 12 wks post-surgery and assessed for PTOA features (for the late *Ccr2* ablation, only the 8 and 12 wk time points were assessed). As controls, we used *CCR2^flx^*^/*flx*^-*eGFP*/*Col1α1ER*-negative mice, in which Cre is not expressed (*CCR2*+/+) (details on genotyping in Supplemental Methods in Appendix B).

### 2.3. Induction of OA

DMM/Sham surgery was induced in the right leg of sixteen-week-old *CCR2*-*Col1αKO* or *CCR2*+/+ male mice by transecting the menisco-tibial ligament, as previously described [19,20,21]. The DMM model results in an injury-induced OA more consistent with the human clinical disorder in that it allows loading during the slow progression of changes in cartilage and bone. DMM lesions progress in stages from early/mild (4 weeks post-surgery) to moderate (8 weeks) and then to severe OA (≥12 weeks). In the Sham, the ligament is visualized and left untouched). For all the experiments performed in this study, only male mice housed in the same room were used as OA severity is markedly higher in males than females after DMM [22] (more details in Appendix B). For each experimental group, Sham and DMM were performed by the same surgeon. Mice subjected to DMM/Sham were assessed for pain behaviors up to 20 wks post-surgery. In a parallel set of experiments, dissected knees from mice at 2, 4, 8, and 12 wks post-surgery were fixed and prepared for histology.

### 2.4. Behavioral Studies

PTOA pain in the DMM model was assessed, as we previously reported [10]. Weight-bearing measures (static spontaneous pain) were obtained by assessing the hindlimb weight distribution with an incapacitance meter, as previously reported [23]. In this test, we measure the force exerted by each limb on a plate in a five-second time window and express it as a ratio: Left/un-operated vs. Right/operated. The ratio is recorded as a pain measure. Higher ratios correlate with increased pain. In addition, the application of von Frey filaments to the hind paws (plantar surface) was used to determine a mechanosensitivity threshold (evoked pain), as previously reported [24]. A mouse responding to a filament of less force (grams) was more sensitive (for more detail, see Appendix B).

### 2.5. Histopathologic Assessment of Arthritis

Histological preparations were obtained, as we previously reported [10]. Briefly, mouse knees were fixed (4% PFA), decalcified (Immunocal, StatLab, McKinney, TX, USA), embedded in paraffin and cut through the entire joint (frontal sections, 6 µm). For each sample, one midcoronal and one posterior sections were stained (hematoxylin and eosin or safranin O/fast green), and pictures were taken using the Olympus BX51 microscope and a DP71 camera. Semiquantitative measures of OA were recorded using the articular cartilage structure score (ACS, scale 0–12) and the Safranin-O staining score (Saf-O, scale 0–12) [25]. The ACS focuses on articular cartilage structure, scoring specifically the fibrillations and clefts in the lamina at the surface of the articular cartilage, while Saf-O is used to identify damage in the extracellular matrix and/or changes in the cell compartment [25] (more details in Appendix B). Adjacent sections were used for H&E and Saf-O staining, and the score is reported as the average of the medial compartment of the joint for each sample (medial tibial plateau and medial femoral condyles).

Histomorphometric analyses using ImageJ were performed to quantify the articular cartilage area, the subchondral bone area and the percentage of bone volume/total volume (BV/TB) in the medial tibia [26] from mice used for semiquantitative assessments [6]. Specifically, for AC quantification, the uncalcified articular cartilage of each slide stained with H&E from all groups was encircled to measure the area. For bone assessment, Saf-O slides from all groups were used. Specifically, the subchondral bone area was defined as the area corresponding to the bone tissue between the calcified cartilage and the trabecular bone that surrounds the bone marrow regions. For BV/TV measurements, we included both the subchondral and trabecular bone, excluding the bone marrow regions (BV) and expressed it as a percentage vs. the total area, including the bone marrow regions (TV). Sections were examined in a blinded fashion with ImageJ software (Available online: http://imagej.nih.gov/ij/ accessed on 14 May 2020) [21]. Data were reported in square micrometers (sq µm) as the average of four sections for each mouse.

### 2.6. Osteophyte Assessment

Semiquantitative measures of osteophyte formation were obtained, as we previously reported [10], using the method described by Little et al. [27]. Briefly, osteophyte size (on a scale of 0–3) was scored by comparison with the thickness of the adjacent articular cartilage; osteophyte maturity (on a scale of 0–3) is used to define the tissue composition of the osteophyte, assigning a lower score to a cartilagineous osteophyte and a higher score to an osteophyte constituted predominantly by bone [27].

As previously reported [10], we quantified the area of cartilaginous tissue (defined by Saf-O staining) within an osteophyte by performing histomorphometric analyses using ImageJ software and following the method of Nagira et al. [26]. Measures were expressed as the ratio between the percentage of Saf-O Area/percentage of osteophyte. Only the medial side of the joint was scored as osteophytes in the DMM model are predominantly localized in this region.

### 2.7. Immunohistochemistry (IHC) Studies

Sections adjacent to those used for histopathologic assessment were used for IHC staining for CCR2 and GFP. Vectastain ABC kit (Vector Laboratories, Burlingame, CA, USA) was used according to the manufacturer’s instructions. Briefly, sections were incubated in 3% H_2_O_2_ diluted with 0.01 M PBS for 10 min, blocked for 1 h with appropriate serum (5%) and incubated in primary antibody (diluted in blocking solution) overnight at room temperature: for anti-CCR2 at the 1:100 dilution for AC, and 1:50 for bone; for anti-GFP at 1:100 for AC, and 1:200 for bone. Following rinses, sections were incubated in the appropriate biotinylated secondary antibody diluted in a blocking buffer (1:200) for 1 h. Sections were then incubated in an Avidin-Biotin complex, and the precipitate was visualized with diaminobenzidine. Time of development changes depending on the tissues analyzed. As a control, representative sections were incubated without primary antibodies to exclude no-specific binding (data not shown).

Images were taken with an Olympus BX51 microscope and a DP71 camera.

### 2.8. Assessment of Synovial Thickness

In order to determine synovial thickness, one H&E-stained section/mouse from the posterior joint compartment was used for a semiquantitative score defined by Rowe et al. (scale of 0–3) [28]. The scale system defines the number of cell layers visualized in the synovium, with higher grades when multiple layers are present (2–3 layers = score 1; 4–5 layers = score 2; 5 or more layers = score 3). Only the posterior slides were scored to avoid potential due to the DMM incision. Measures are expressed as an average of the medial and lateral compartments of the joint.

All parameters were measured by three independent investigators in a blinded manner, and the results were expressed as an average.

### 2.9. Statistical Analysis

Statistical analysis was completed using SAS v 9.4 (SAS Institute), and significance was set at 0.05. The normality of data and homogeneity of variance were determined by Shapiro–Wilk and Levene’s tests. Most outcomes (ACS, Saf-O score, and all the histomorphometric analyses) violated the normality assumption. Therefore, we used non-parametric Wilcoxon rank sum tests following Benjamini and Hochberg’s adjustment for *p*-values in multiple comparisons. Each experimental time point was analyzed separately.

We used Wilcoxon tests to score osteophyte size, osteophyte maturity and synovial thickness.

For behavioral pain measures, to account for longitudinal measures over time, we used linear mixed-effects models fit with unstructured covariance structure. Fixed-effect terms comprised group indicators (Sham or DMM for different genotypes, as well as left and right for von Frey analyses), time points, and how they interact. In Von Frey’s analyses, the model included a random mouse effect to account for measurements on both knees (operated and contralateral). Q-Q plot was used to check the model assumption of normality of residuals. A plot of residuals vs. fitted values was used to check the equal variance of residuals. A square root transformation was applied to reach normality in all Von Frey measurements. Missing measurements were still included in the model (‘missing at random’ paradigm). We used Proc plm to estimate comparisons, specifying differences between means. GraphPad Prism Software (9.1.0) was used for data plots.

## 3. Results

### 3.1. Ccr2 Inactivation in Bone Tissue

We successfully generated inducible *CCR2*-*Col1αKO* mice by injecting tamoxifen in *CCR2^flx^*^/*flx*^-*eGFP*/*Col1αCreER*-positive mice, where Cre recombination leads to Gfp expression. Injected *CCR2^flx^*^/*flx*^-*eGFP*/*Col1αCreER*-negative mice were used as controls (*CCR2*+/+). Figure 1A shows the presence of GFP protein staining in the bone tissue of *CCR2*-*Col1αKO* two weeks after the first injection, while it was undetectable in the *CCR2*+/+. To confirm the CCR2 inactivation, we assessed protein levels of CCR2 in the bone tissue and, as expected, we found CCR2 staining in the bone of *CCR2*+/+ mice but not in *CCR2*-*Col1αKO* (Figure 1B). As a control, we evaluated GFP and CCR2 levels in the cartilage compartment and, as expected, we confirmed the presence of CCR2 staining and the absence of GFP in both genotyping, confirming Cre specificity in the bone compartment (Appendix A).

### 3.2. Early Osteoblast-Ccr2 Inactivation Decreases Cartilage Damage Induced by Injury

To determine the contribution of *Ccr2* osteoblast expression to PTOA onset, we induced Cre recombination in *CCR2^flx^*^/*flx*^-*eGFP*/*Col1CreER^T2^*-positive mice before DMM to obtain *CCR2*-*Col1αKO* and followed PTOA damage in cartilage, bone, and synovium at mild (two, four weeks), moderate (eight weeks), and severe (twelve weeks) stages. For the cartilage damage, we used both the ACS and Saf-O semiquantitative scores in order to provide a more comprehensive characterization of the lesions [25]. In Figure 2A,B, the histology visualizes a complete degeneration of the articular cartilage across all PTOA stages; in particular, the absence of articular cartilage (AC in Panel B) in *CCR2*+/+ mice at the most severe stage (12 weeks), while the *CCR2*-*Col1αKO* littermate has a less severe phenotype. This outcome is translated in the semiquantitative ACS reported in Panel C, where early *Ccr2* inactivation leads to a decreased score compared to the *CCR2*+/+. Of note, although differences were mostly detected at the severe stage, some improvement was also detected at the very early stages (two weeks post-surgery), while differences were not detected at the middle stages (four to eight weeks post-surgery) (Table 1).

Mirroring the ACS score, DMM *CCR2*-*Col1αKO* mice showed improvement of the extracellular matrix both early after surgery and at the severe PTOA stages, while differences were not detected between compared to DMM *CCR2*+/+ (Figure 2D,E, Table 1). We can see that DMM *CCR2*+/+ mice progressively lose the Saf-O staining (red color in Panels D,E), which becomes almost undetectable at the most severe stage (12 weeks). As for ACS, the DMM *CCR2*-*Col1αKO* littermates preserve some staining, which is reported as a Saf-O score in Figure 2F. We also performed AC histomorphometric analysis to quantify the thinning of the articular cartilage induced by DMM. As shown in Figure 2G and Table 1, DMM *CCR2*-*Col1αKO* mice showed increased AC area compared to DMM *CCR2*+/+ only at the severe stages. When sample means were plotted and analyzed within the same genotypes (Appendix A), it is evident that *Ccr2* osteoblast ablation slows the progression of injury-induced cartilage damage mostly between eight and twelve weeks for all cartilage parameters analyzed (ACS, Saf-O and Cartilage Quantification). In contrast, DMM *CCR2*+/+ mice show an increase in cartilage damage scores up to 12 weeks post-injury (Appendix A). No differences were detected in all Sham surgeries at all time points. Therefore, Shams were not reported.

### 3.3. Early Osteoblast-Ccr2 Inactivation before Injury Reduces Bone Thickness but Does Not Affect Osteophyte Formation

We evaluated changes in bone tissues in *CCR2*-*Col1αKO* and *CCR2*+/+ littermates after early *Ccr2* inactivation, such as osteophyte formation, subchondral bone thickness in the medial tibia, as well as measured the total bone volume comprised between the calcified cartilage and the growth plate. We rarely observe osteophytes in Shams at all time points. Therefore, we excluded Shams from the analysis. As shown in Figure 3A,B and Table 2, early osteoblast-*Ccr2* deletion does not affect osteophyte size during OA progression. Similarly, the osteophyte tissue composition measured as maturity score (Figure 3C) and osteophyte cartilage quantification (Figure 3D) were unchanged at all time points. When evaluating subchondral bone thickness and the percentage of total bone, we found that Col1α-specific *Ccr2* ablation led to extensive bone protection, decreasing both the DMM-induced subchondral thickness (Figure 3E) as well as the percentage of bone volume vs. total volume (Figure 3F,G). Similarly to cartilage changes, early *Ccr2* osteoblast ablation is most effective on late bone changes, where it is more evident that the damage slows its progression (Appendix A). No changes in subchondral bone thickness were assessed in all Shams (data not shown). Therefore, they were not included in the analysis.

### 3.4. Early Osteoblast-Ccr2 Inactivation Does Not Affect Synovial Hyperplasia Induced by Injury

CCR2 is a chemokine receptor involved in recruiting macrophages to the site of inflammation [1,2]. Therefore, we analyzed whether genetic ablation of *Ccr2* in osteoblast was affecting the DMM-induced synovial hyperplasia. We did not find any differences among different genotypes, as shown in Appendix A. Synovial thickness is rarely present in Shams. Therefore, Shams were not included.

### 3.5. Late Osteoblast-Ccr2 Inactivation during OA Progression Decreases Cartilage Damage Induced by DMM

To determine how osteoblast levels of *Ccr2* affect PTOA progression after the initial onset, we induced Cre recombination in *CCR2^flx^*^/*flx*^-*eGFP*/*Col1αCreER*-positive mice four weeks after DMM surgery and followed joint degeneration in *CCR2*-*Col1αKO* and *CCR2*+/+ mice at moderate and severe stages (eight to twelve weeks). As shown in Figure 4A–C and Table 3, late *Ccr2* inactivation was able to slow the damage on the surface of the articular cartilage (ACS score) at the severe stage. Similar results were found on extracellular matrix composition, where late Cre recombination decreased the Saf-O score (Figure 4D–F) at the 12-week time point. Improvement in AC area quantification was also detected in the *CCR2*-*Col1αKO* compared to *CCR2*+/+ mice at the severe stage (Figure 4G and Table 3). Similarly to early recombination, when samples are plotted and analyzed within each genotype, there is no noticeable progression in the cartilage damage of *CCR2*-*Col1αKO* mice between eight weeks and twelve weeks, while the *CCR2*+/+ littermates show an increase in the cartilage degeneration. (Appendix A).

### 3.6. Late Osteoblast-Ccr2 Inactivation during OA Progression Delays Bone Thickness but Does Not Affect Osteophyte Formation

We next evaluated the DMM-induced bone changes in *CCR2*-*Col1αKO* and *CCR2*+/+ mice following late *Ccr2* inactivation. As for the early *Ccr2* inactivation, late *Ccr2* deletion during OA progression did not affect osteophyte size at any disease stages (Figure 5A,B and Table 4). In contrast to the early ablation, a difference in osteophyte tissue composition was detected between different genotypes at the PTOA severe stage (12 weeks), detected as a lower maturity score in the *CCR2*-*Col1αKO* (less bone, Figure 5C) and confirmed by a higher percentage of cartilage quantification (Figure 5D). As mentioned above, Shams were excluded from the analysis. Late *Ccr2* inactivation decreased the DMM-induced subchondral thickness (Figure 5E, Table 4) and decreased the percentage of Bone Volume (%BV/TV, Figure 5F,G, Table 4) at both middle (eight weeks) and severe (twelve weeks) PTOA stages although data reached statistical significance only at the most severe time point.; subchondral bone thickness seemed to slow progression in the *CCR2*-*Col1αKO* between eight and twelve weeks, while no changes were evident in the %BV/TV. However, changes were minimal, and data were not statistically significant (Appendix A).

### 3.7. Late Osteoblast-Ccr2 Inactivation Does Not Affect Synovial Hyperplasia Induced by Injury

Similar to early osteoblast-*Ccr2* inactivation, no changes in synovial hyperplasia score were detected between *CCR2*-*Col1αKO* and *CCR2*+/+ mice at the time points observed (eight and twelve weeks), as shown in Appendix A.

### 3.8. Reduced PTOA Joint Damage following Osteoblast-CCR2 Inactivation Diminishes Pain Responses at the Severe Stage

In previous studies, we demonstrated that spontaneous pain behavior in mice could be reduced by early blockage of CCR2 signaling using a chemical receptor antagonist or by early deletion of the *Ccr2* gene in chondrocytes [6,10]. In this study, we analyzed whether the ablation of the *Ccr2* gene in osteoblasts and the consequent decrease in joint damage mirrored a lower susceptibility to pain compared to mice carrying the intact *Ccr2* gene. Our data demonstrated that ablation of *Ccr2* in osteoblasts improved spontaneous pain compared to *CCR2*+/+ mice, measured by hindlimb weight distribution.

Early osteoblast-*Ccr2* deletion in DMM mice resulted in decreased pain responses during PTOA progression, compared to DMM-induced *CCR2*+/+ (Figure 6A, DMM *CCR2*+/+ vs. DMM *CCR2*-*Col1αKO*, yellow line vs. red line, respectively). Table 5 includes estimated between-group differences (corresponding 95% CIs) at each time point. In particular, as OA progressed up to 20 weeks, *CCR2*-*Col1αKO* injured mice had values similar to their Sham *CCR2*-*Col1αKO* littermate controls (Figure 6A and Appendix A DMM *CCR2*-*Col1αKO* vs. Sham *CCR2*-*Col1αKO*, red line vs. purple line, respectively). In contrast, DMM *CCR2*+/+ mice progressed vs. higher static pain measures, as compared to Sham *CCR2*+/+ mice (Figure 6A and Appendix A, DMM *CCR2*+/+ vs. Sham *CCR2*+/+, yellow line vs. green line, respectively).

We also evaluated changes in evoked pain caused by DMM using von Frey filaments Figure 6B, Table 5 and Appendix A). We found differences in pain values between the DMM leg vs. the contralateral leg in both genotypes (starting at 12 weeks); however, when comparing DMM legs between *CCR2*-*Col1αKO* and *CCR2*+/+ mice, no differences were detected at any time point. No differences were detected among all the Shams of both genotypes, including operated and contralateral legs (data not shown).

## 4. Discussion

OA is considered a disease of the whole joint, affecting not only the articular cartilage but extending to the bone tissue and synovium. Bone damage in PTOA, such as the thickening of the area below the calcified cartilage (subchondral bone and trabecular bone) and osteophyte formation, can appear early during disease progression, and such structural changes can contribute to OA pain and disability [6,9,29,30]. Therefore, in the effort to find therapeutic intervention to limit OA progression, bone tissue constitutes an important target.

Human studies correlating *Ccr2* and *Ccl2* gene variants with osteopenia and osteoporosis risk demonstrated the important role of the CCR2 pathway in the skeletal system [31]. In accordance, several animal studies in *Ccr2 null* mice confirmed that lack of *Ccr2* associated with impaired osteoclastogenesis and bone resorption (reviewed by Zhu et al. 2021) [32] and also suggested a direct role of CCR2 in the osteoblast niche, which could affect osteoblast differentiation and regulate bone growth in pathological conditions [6,14,15]. However, the pre-and-post-natal growth plate defects associated with global germline *Ccr2* deletion in the *Ccr2 null* mice render this model not ideal for studying OA development [10]. In previous PTOA mice studies, we found increased protein levels of CCR2 in chondrocytes, osteoblasts, and synovium early after injury and have used a specific CCR2 antagonist (RS504393) to inhibit the CCR2 signaling in adult mice, contributing to our understanding of the role of this receptor during DMM-induced OA. We reported a protective action on cartilage and bone structure by a systemic delivery at the time of the injury for a limited amount of time (three weeks) [6], as well as by an intraarticular slow release in the injured knee for ten weeks using a microplate-based drug delivery system [9]. However, this pharmacological approach does not reveal the contribution of tissue-specific levels of CCR2 to the whole joint pathology. In a recent study, we used an inducible aggrecan-specific *Ccr2KO* mouse model to study PTOA in the absence of cartilage levels of *Ccr2*; using this model, we validated the PTOA preventive action of CCR2 targeting on cartilage damage but detected a limited efficacy on bone changes [10], suggesting that levels of *Ccr2* in other cell compartments could directly contribute to PTOA bone changes.

To determine how *Ccr2* deletion in the bone compartment could impact the whole joint pathology, in this study, we induced *Ccr2* deletion in osteoblast cells, modulating its ablation after skeletal maturity, either before injury (early ablation) or during the course of PTOA (late ablation). It is known that bone-cartilage crosstalk contributes to joint homeostasis. Therefore, osteoblast levels of *Ccr2* could indirectly affect articular cartilage structure; we found that both early and late *Ccr2* ablation affected cartilage degeneration, although the amelioration was more efficient at the severe OA stage (Figure 2, Figure 3 and Figure 4; Appendix A). These results differ from previous data obtained with chondrocyte-specific *Ccr2* ablation, where only an early approach was successful in preventing cartilage damage, with high efficiency on early OA [10], while later intervention resulted in lost efficacy. If these results are taken together, they suggest that although *Ccr2* expression in both chondrocytes and osteoblasts contributes to cartilage degeneration, the expression in the two cell compartments might regulate cartilage degeneration at different stages during disease progression, with the osteoblast niche becoming more critical when the disease is already established. This could be an indirect consequence of the more severe bone changes occurring at end-stage OA. Our data on bone damage following osteoblast-*Ccr2* inactivation appear to corroborate this hypothesis.

The subchondral bone is the region lying beneath the calcified cartilage, and its integrity is critical in OA [33]; the subchondral bone is a dynamic structure that not only provides support for the mechanical forces that insist on the joint structure [34] but interacts with the articular cartilage to ensure a correct nutrient supply and metabolism [35], contributing to joint homeostasis [33]. Following DMM, the subchondral bone of the medial side of the knee becomes denser as a consequence of the injury, with differences more evident between five and ten weeks post-surgery [36]. We found that the bone thickening of the medial tibia normally induced by the ligament injury was significantly reduced in *CCR2*-*Col1αKO* mice when compared to *CCR2*+/+ littermates. As for cartilage damage, this improvement was detected only at the most severe stage, either with early or late *Ccr2* inactivation (Figure 3E–G and Figure 5E–G; Appendix A). Because no differences in osteoclast activity were detected between the two genotypes (data not shown), our data suggest that *Ccr2* expression in the osteoblast population is critical to drive the bone thickness in both the subchondral and trabecular compartments, and such changes might partially influence the integrity of the cartilage above.

When we compared all the PTOA outcomes between early and late inactivation, we found no significant differences in phenotypes at both the middle and severe stages (Appendix A); these results corroborate the hypothesis that bone expression levels of *Ccr2* are more critical on OA progression after the disease is established and suggest that the cartilage degeneration might be an indirect action of the bone thickening.

We also analyzed osteophyte formation in the absence of osteoblast-*Ccr2*. In the DMM, osteophyte formation begins as a consequence of trauma as early as two weeks post-DMM [6,36] (Figure 3A). Unexpectedly, osteoblast *Ccr2* ablation was not sufficient to decrease osteophyte formation during DMM-induced OA, although some efficacy was noticed in their maturation from cartilage to bone (Figure 3C,D and Figure 5C,D). These results mirror our data obtained previously with chondrocyte-specific *Ccr2* inactivation, suggesting that *Ccr2* expression in both the cartilage and bone compartments is not the main driver of osteophyte formation, although it might contribute to their maturation from cartilage to bone [10]. Importantly, our previous data with pharmacological CCR2 targeting successfully demonstrated the role of CCR2 activation in the process of osteophyte formation, suggesting that other target cells might be critical for the process. In this respect, Fang et al. deeply analyzed early changes in the bone compartment of DMM knees and reported differences in osteoclast activity, mostly localized at the site of osteophyte formation, although some change was also seen in the subchondral bone [36]. This suggests that osteoclast expression levels of *Ccr2* might contribute to the process of osteophyte formation in the DMM knees. We need to acknowledge that our experimental method to analyze bone damage relies on the sole histological evaluation of tissues, as reported by Nagira et al. [26]; although this method has been specifically validated for the evaluation of bone changes in OA murine models by comparing it with µCT analyses, more sensitive techniques would allow a more accurate interpretation of bone parameters, and this constitutes a limitation of the study. In this respect, further studies on larger animals, coupling more translational methods for CCR2 targeting with more sensitive experimental techniques (such as µCT and MRI), will be needed to evaluate the potential application of CCR2 targeting in OA.

We also evaluated whether *Ccr2* deletion in osteoblasts affected the synovial hyperplasia induced by DMM. As reported for chondrocyte-specific inactivation, no differences were detected between genotypes with early or late inactivation at any of the PTOA stages analyzed (Appendix A). Interestingly, previous data with intraarticular pharmacological CCR2 inactivation demonstrated a decreased synovial thickness following injury. When taken together, our results suggest that the *Ccr2* expression in the osteoclast/macrophage niche might play a critical role in PTOA pathogenesis, altering bone remodeling in some joint compartments and affecting inflammation. Further studies are needed to understand the role of *Ccr2* in other cell populations in the contest of PTOA.

Pain can originate from many joint tissues [37], and it is the main reason for disability in the OA population [38,39]. Although cartilage degeneration and synovial inflammation are believed to be linked to OA pain, different studies have reported an important role for subchondral bone as a source of pain in OA [40,41]. The formation of new bone and osteophytes is normally accompanied by angiogenesis and consequent innervation and, therefore, can affect pain perception during OA progression [42]. The important role of CCR2 in pain has been confirmed by different investigators, including us. We recently established a link between expression levels of the *Ccr2* gene in cartilage with joint tissue degeneration post-trauma and pain [10]. Interestingly, *Ccr2* ablation in osteoblasts led to very similar outcomes, with static pain that was ameliorated in *CCR2*-*Col1αKO* mice vs. *CCR2*+/+ littermates, although the effect was noticeable at later stages compared to chondrocyte inactivation (16 weeks post-trauma vs. 12 weeks) [10]. This is not surprising, given the fact that *Ccr2* bone levels seem to mediate joint damage at later stages compared to *Ccr2* chondrocyte levels [10]. In contrast to static pain, evoked pain by Von Frey filaments showed some improvements, but differences among genotypes did not reach statistical significance at any of the time points analyzed, suggesting that different mechanisms might regulate evoked vs. spontaneous pain. Our result mirrors a human knee osteoarthritis study that revealed a different brain activity in evoked pain versus spontaneous pain [43]. A study using *Ccr2* null mice reported that CCR2 had a different contribution to chronic pain vs. acute pain [44]. Other studies have established a correlation between specific tissue damage with joint pain [45,46]. For example, osteophyte formation seems to be less critical for pain perception than joint space narrowing, probably because the presence of osteophytes might have some beneficial effect in stabilizing the joint [46]. Therefore, in our case, the difference between evoked and spontaneous pain might reflect particular structural changes in bone or cartilage tissues; however, because bone changes were indirectly affecting cartilage damage, it was impossible to define the tissue source of pain. It is important to remark that evoked mechanosensitivity in the absence of OA degeneration (contralateral legs of both *CCR2*-*Col1αKO* and *CCR2*+/+) showed similar values along all-time points (Appendix A), confirming that the ablation of *Ccr2* in osteoblasts alone does not translate in changes in mechanosensitivity in the absence of joint tissue degeneration.

There is no treatment for OA, and the multi-tissue nature of the pathology renders it more difficult to find a successful target to halt or slow disease progression. CCR2 and its ligand CCL2 are expressed in macrophages, neurons, chondrocytes, osteoblasts and tendon fibroblasts [6]. Thus, the relevance of the CCLs/CCR2 axis for potential PTOA therapies is not limited to mediating the inflammatory response accompanying disease progression [4,5,6], but differently from other chemokines, is directly linked to cartilage and bone integrity [2,6,9,10], representing a potential target also for tissue structure and pain [47]. However, before translating to the clinic, it is critical to understand the contribution of the CCL2/CCR2 axis to each tissue affected by OA and how each tissue contributes to the whole pathogenesis. Our study provides important insights into potential CCR2-targeted therapies for OA. by defining how Ccr2 levels in the bone compartment affect the whole joint structure and pain at different disease stages.

## Figures and Tables

**Figure 1 biomolecules-13-00891-f001:**
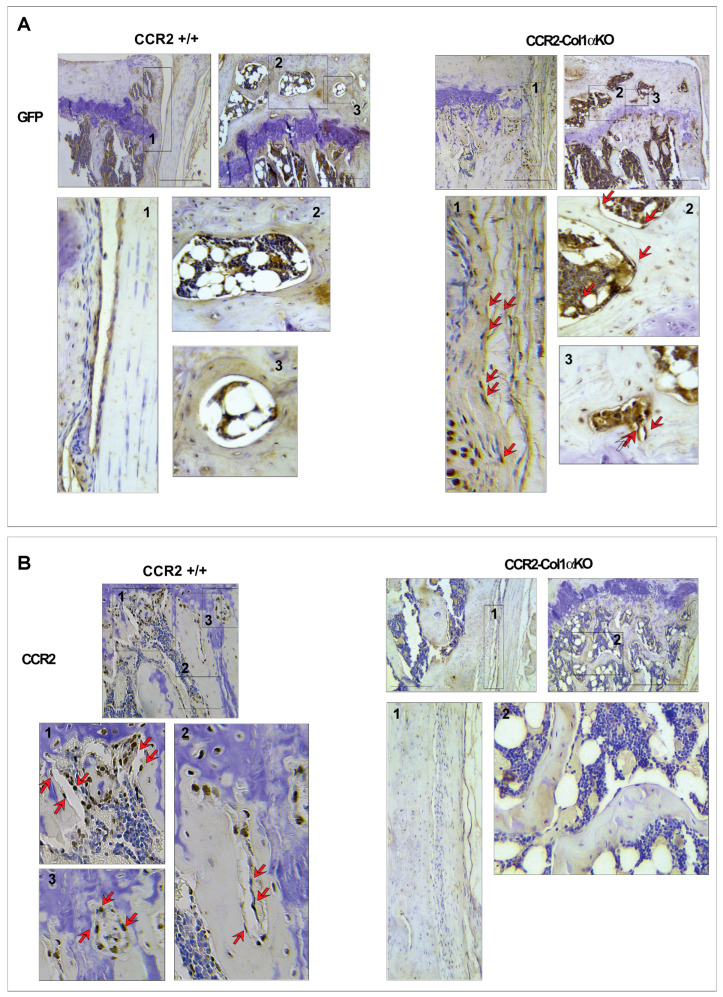
**Protein levels of CCR2 and GFP in the bone tissue of *CCR2*-*Col1αKO* and *CCR2*+/+ mice following Tamoxifen injection.** Paraffin-embedded knee joint sections are immunostained for GFP (**A**) and CCR2 (**B**) two weeks after the first Tam injection. Positive staining is detected as a brown precipitate (red arrows). GFP staining is visible in *CCR2*-*Col1αKO* (red arrows highlight a few brown nuclei) but undetected in *CCR2*+/+ mice (blue nuclei), indicating recombination. Conversely, CCR2 staining is detected in the bone cells of *CCR2*+/+ mice, while it is absent in *CCR2*-*Col1αKO*. Images in the rectangles represent a magnification of specific numbered regions in the main picture. Images are representative of 6 different mice for each of the experimental groups described, ranging between 14 and 18 weeks of age. Scale bars are 100 µm.

**Figure 2 biomolecules-13-00891-f002:**
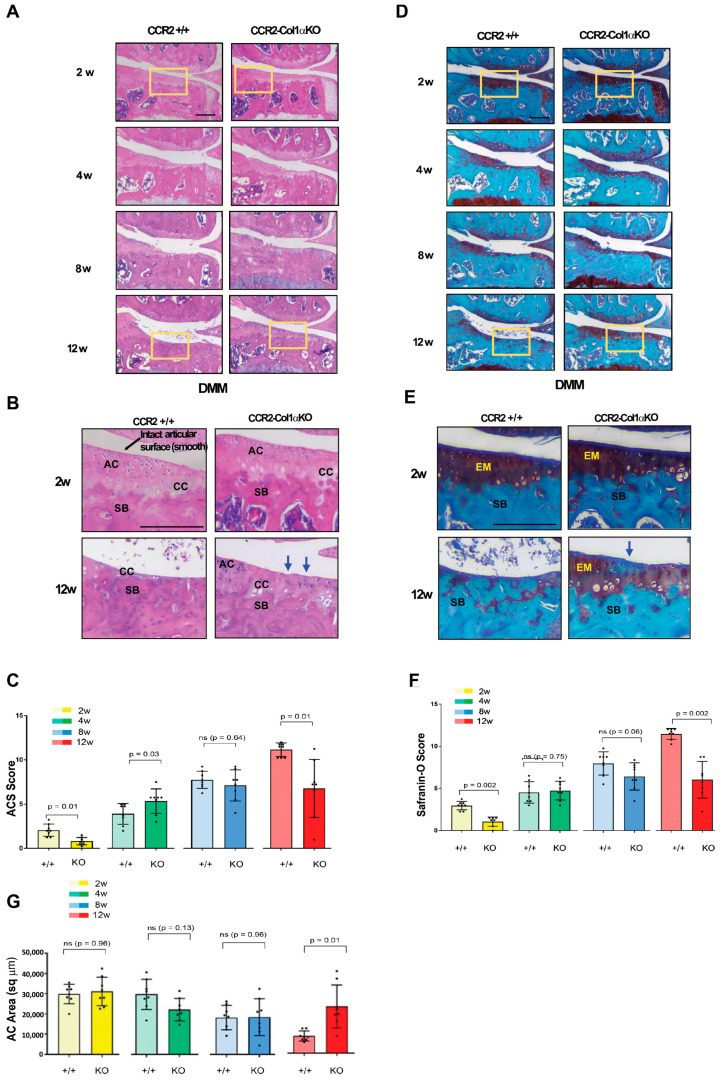
**Histopathological measures of cartilage damage in DMM mouse knees following early *Ccr2* inactivation.** (**A**) Panel A shows H&E staining of the medial knee compartment of reported genotypes following DMM at the indicated time points. (**B**) Panel B shows magnified images corresponding to the squares in panel A (AC, articular cartilage; CC, calcified cartilage; SB, subchondral bone). The blue arrows indicate regions of lesions in the articular cartilage). (**C**) Panel C shows the ACS score (scale 0–12) of knees subjected to DMM at the indicated time points. Reported measures represent an average of the medial joint compartment (medial tibia and medial femurs), as this is the most affected area. (**D**) Panel D shows Safranin-O/Fast green staining of the medial compartment of mouse knees following DMM at the indicated time points. (**E**) Panel E shows magnified images corresponding to the squares in panel D (EM, extracellular matrix; SB, subchondral bone). The blue arrow indicates bone tissue on the joint surface. (**F**) Panel F shows the Saf-O score (scale 0–12) of knees subjected to DMM at the indicated time points. Reported measures represent an average of the medial joint compartment (medial tibia and medial femurs). (**G**) Panel G shows measures of articular cartilage area (sq µm) at the indicated time points, quantified by histomorphometric analyses (medial compartment). Reported images are representative of N = 8 for each time point; scale bars in the images are set at 100 µm. The plots represent the mean ± standard deviation. Wilcoxon rank sum tests were used to calculate *p*-values at each time point, following adjustment for multiple comparisons (ns = not significant).

**Figure 3 biomolecules-13-00891-f003:**
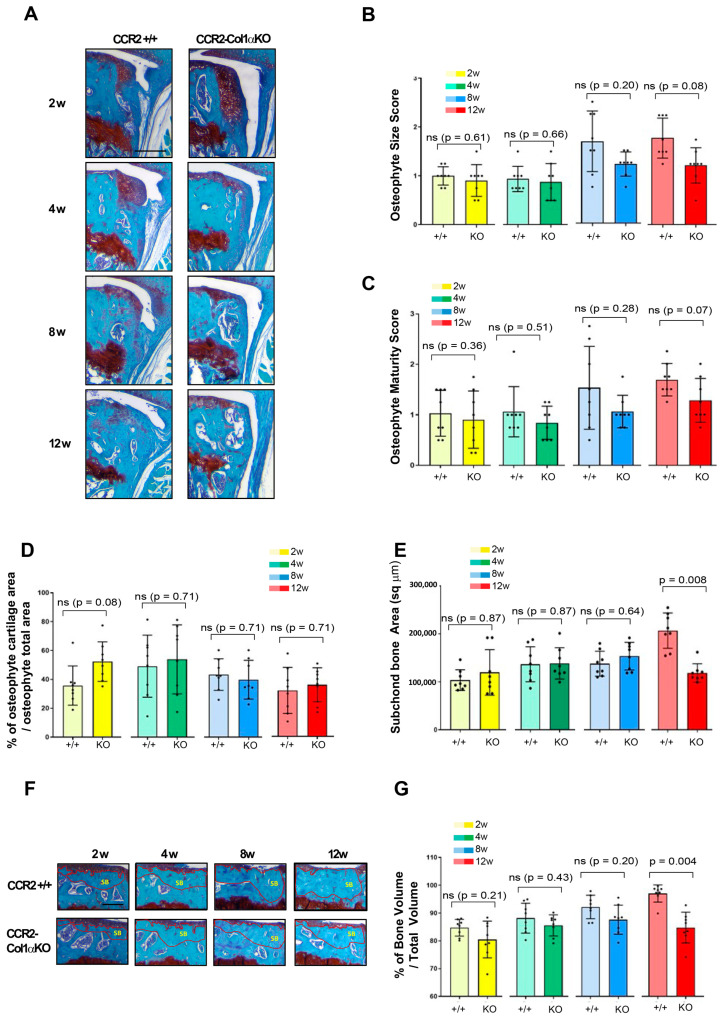
Assessment of DMM-induced bone damage in mouse knee joints of *CCR2*-*Col1αKO* and *CCR2*+/+, following early *Ccr2* inactivation. (**A**) Safranin-O/Fast green staining of osteophytes in the medial compartment of *CCR2*-*Col1αKO* and *CCR2*+/+ mouse knees (tibia) at two, four, eight, and twelve weeks after DMM surgery. (**B**) Osteophyte size and (**C**) osteophyte maturity scores (scale 0–3) of *CCR2*-*Col1αKO* and *CCR2*+/+ mouse knees at the time point indicated and described in panel A. Results are expressed as the average of four quadrants (medial and lateral tibia, medial and lateral femurs). (**D**) Percentage of cartilage tissue (sq µm) present in the osteophytes of DMM knees (only the bigger osteophyte among the four quadrants) by histomorphometric analysis at the time point indicated. (**E**) Quantification of the subchondral bone area (sq µm) of the medial plateau of DMM knees by histomorphometric analysis at the time point indicated (visualized in Figure 3F). (**F**) Safranin-O/Fast green staining of the medial bone compartment comprised between the AC and growth plate (defined as Total Volume, blue color) of *CCR2*-*Col1αKO* and *CCR2*+/+ mouse knees at two, four, eight, and twelve weeks after DMM surgery. The subchondral bone area of each representative section has been circled. (**G**) Percentage of bone tissue (Bone Volume, BV) vs. the Total Volume (TV, defined as in panel F) of DMM knees (medial compartment) by histomorphometric analysis at the time point indicated (sq µm). All images are representative of N = 8 for each of the experimental points described; scale bars are 100 µm. The graphs represent the mean ± standard deviation (N = 8); indicated *p*-values were determined by Wilcoxon rank sum tests at each time point, following adjustment for multiple comparisons (ns = not significant).

**Figure 4 biomolecules-13-00891-f004:**
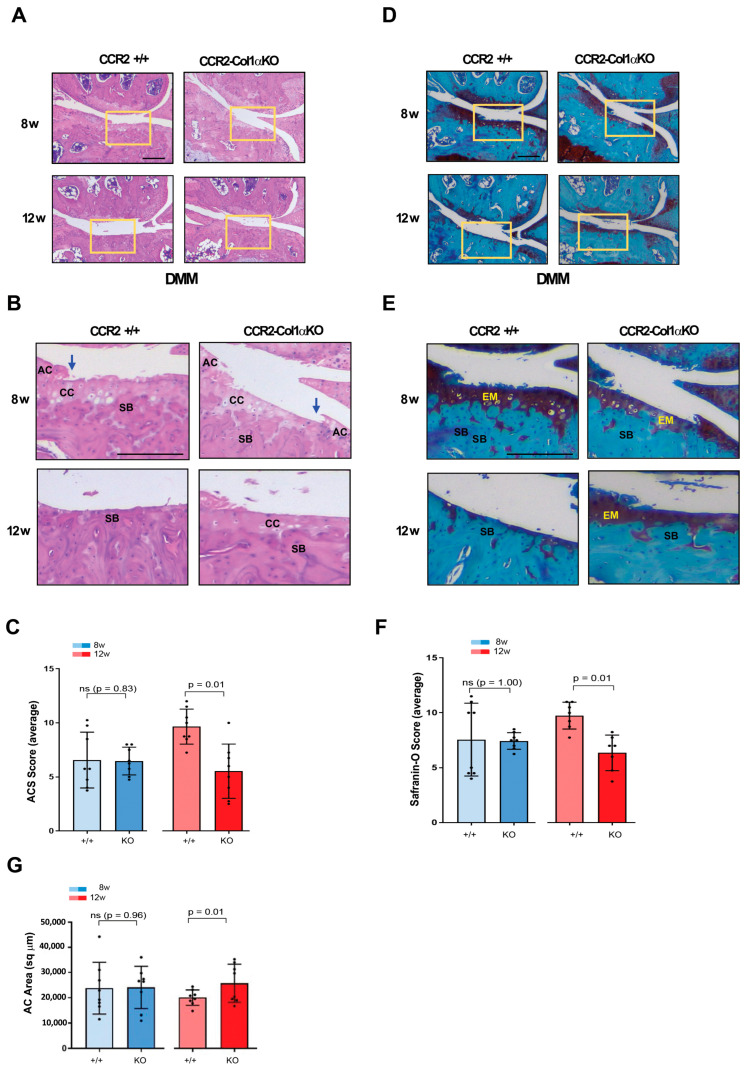
**Histopathological measures of cartilage damage in DMM mouse knees following late *Ccr2* inactivation.** (**A**) Panel A shows H&E staining of the medial knee compartment of reported genotypes following DMM at the indicated time points. (**B**) Panel B shows magnified images corresponding to the squares in panel A (AC, articular cartilage; CC, calcified cartilage; SB, subchondral bone). The blue arrows indicate regions of lesions in the articular cartilage). (**C**) Panel C shows the ACS score (scale 0–12) of knees subjected to DMM at the indicated time points. Reported measures represent an average of the medial joint compartments (medial tibia and medial femurs), as this is the most affected area. (**D**) Panel D shows Safranin-O/Fast green staining of the medial compartment of mouse knees following DMM at the indicated time points. (**E**) Panel E shows magnified images corresponding to the squares in panel D (EM, extracellular matrix; SB, subchondral bone). (**F**) Panel F shows the Saf-O score (scale 0–12) of knees subjected to DMM at the indicated time points. Reported measures represent an average of the medial joint compartments (medial tibia and medial femurs). (**G**) Panel G shows measures of articular cartilage area (sq µm) at the indicated time points, quantified by histomorphometric analyses (medial compartment)). Reported images are representative of N = 8 for each time point; scale bars in the images are set at 100 µm. The plots represent the mean ± standard deviation. Wilcoxon rank sum tests were used to calculate *p*-values at each time point, following adjustment for multiple comparisons (ns = not significant).

**Figure 5 biomolecules-13-00891-f005:**
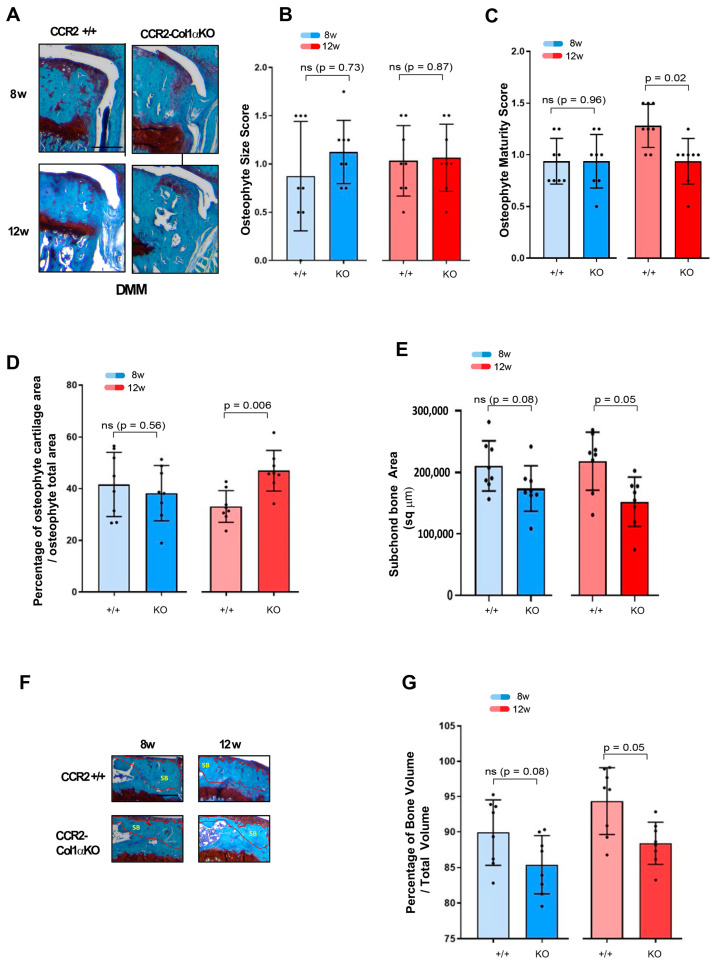
Assessment of DMM-induced bone damage in mouse knee joints of *CCR2*-*Col1αKO* and *CCR2*+/*+,* following late *Ccr2* inactivation. (**A**) Safranin-O/Fast green staining of the osteophytes in the medial compartment of *CCR2*-*Col1αKO* and *CCR2*+/+ mouse knees (tibia) at eight and twelve weeks after DMM surgery. (**B**) Osteophyte size and (**C**) osteophyte maturity scores (scale 0–3) of *CCR2*-*Col1αKO* and *CCR2*+/+ mouse knees at the time point indicated and described in panel A. Results are expressed as the average of four quadrants (medial and lateral tibia, medial and lateral femurs). (**D**) Percentage of cartilage tissue (sq µm) present in the osteophytes of DMM knees (the larger osteophyte among the four quadrants) by histomorphometric analysis at the time point indicated. (**E**) Quantification of the subchondral bone area (sq µm) of the medial plateau of DMM knees by histomorphometric analysis at the time point indicated (visualized in Figure 5F). (**F**) Safranin-O/Fast green staining of the medial bone compartment comprised between the AC and growth plate (defined as Total Volume) of *CCR2*-*Col1αKO* and *CCR2*+/+ mouse knees at eight and twelve weeks after DMM surgery. The subchondral bone area of each representative section has been circled. (**G**) Percentage of bone tissue (Bone Volume, BV) vs. the Total Volume (TV, defined as in panel F) of DMM knees (medial compartment) by histomorphometric analysis at the time point indicated (sq µm). All images are representative of N = 8 for each of the experimental points described; scale bars are 100 µm. The graphs represent the mean ± standard deviation (N = 8); indicated *p*-values were determined by Wilcoxon rank sum tests at each time point, following adjustment for multiple comparisons (ns = not significant).

**Figure 6 biomolecules-13-00891-f006:**
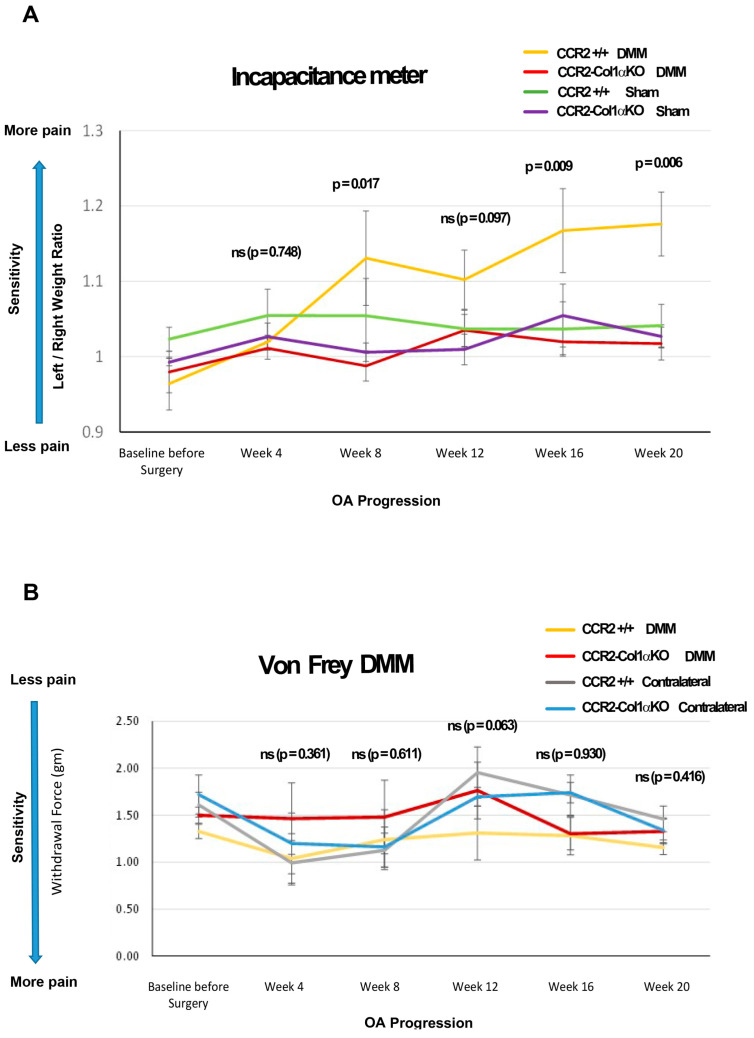
**Pain behavioral measures in DMM mice following CCR2 inactivation.** (**A**) Panel A shows static measures of pain assessed by an Incapacitance Meter in DMM and Sham mice in both genotypes at the time points indicated (N = 8). (**B**) Panel B shows longitudinal measures of evoked pain by von Frey analyses in both genotypes, in both the operated (DMM leg) and the Contralateral limb, at the time points indicated (N = 8). For both graphs, baseline values were obtained before *Ccr2* recombination and prior to surgery. The observed mean scores in each experimental group are plotted at each time point, with error bars indicating the SEM. *p*-values for group differences were calculated by linear mixed effects model (LMM) as described in the Methods (ns = not significant); in particular, the *p*-values in both Panel A and B refer to differences between DMM *CCR2*-*Col1αKO* vs. DMM *CCR2*+/+ mice (yellow line vs. red line); numerical LMM values for differences among other groups are shown in Table 3 and Appendix A (for the Incapacitance Meter’s measures in Panel A) and Appendix A (for the von Frey’s measures in Panel B).

**Table 1 biomolecules-13-00891-t001:** Effect of early *Ccr2* inactivation (before DMM) on cartilage damage in mouse *CCR2*-*Col1αKO* and *CCR2*+/+ knee joints (N = 8).

**OA Parameters**		**Week 2**				**Week 4**			
		**Z (*p*-Value)**	**Mean Score** **(Rank Sums)**	**Median**	**Mean (SD)**	**Z (*p*-Value)**	**Mean Score** **(Rank Sums)**	**Median**	**Mean (SD)**
ACS	DMM *CCR2*+/+	3.01 (0.01)	12.13	2	2.06 (0.69)	−2.23 [0.03]	5.81	4.25	3.91 (1.17)
	DMM *CCR2*-*AggKO*		4.88	0.75	0.81 (0.42)		11.19	5.75	5.34 (1.38)
Saf-O	DMM *CCR2*+/+	3.34 (0.002)	12.5	3	2.97 (0.51)	−0.32 [0.75]	8.06	4.25	4.53 (1.27)
	DMM *CCR2*-*AggKO*		4.5	1.25	1.06 (0.55)		8.94	4.75	4.75 (1.11)
Cartilage Histomorphometry	DMM *CCR2*+/+	−0.05 (0.96)	8.38	30,842	29,817 (4810)	1.84 (0.13)	10.75	29,506	29,551 (7452)
	DMM *CCR2*-*AggKO*		8.63	29,873	31,073 (7051)		6.25	21,135	21,992 (5516)
**OA Parameters**		**Week 8**				**Week 12**			
		**Z (*p*-Value)**	**Mean Score** **(Rank Sums)**	**Median**	**Mean (SD)**	**Z (*p*-Value)**	**Mean Score** **(Rank Sums)**	**Median**	**Mean (SD)**
ACS	DMM *CCR2*+/+	0.47 (0.64)	9.13	7.63	7.75 (0.97)	3.01 (0.01)	12.13	11.5	11.16 (0.77)
	DMM *CCR2*-*AggKO*		7.88	7.5	7.13 (1.76)		4.88	7.25	6.78 (3.29)
Saf-O	DMM *CCR2*+/+	2.00 (0.06)	10.94	8.38	7.97 (1.38)	3.33 (0.002)	12.5	11.63	11.47 (0.65)
	DMM *CCR2*-*AggKO*		6.06	6.63	6.42 (1.62)		4.5	6.13	6.03 (2.18)
Cartilage Histomorphometry	DMM *CCR2*+/+	−0.05 (0.96)	8.38	18,963	18,161 (6042)	−2.99 (0.01)	4.88	7835	8369 (2508)
	DMM *CCR2*-*AggKO*		8.63	18,218	18,364 (9115)		12.13	20,436	22,937 (10,655)

Data presented show the median, mean, and standard deviation (SD) for each group at separate time points (two, four, eight, and twelve weeks post-surgery). Indicated *z*- and *p*-values were determined by Wilcoxon rank sum tests between DMM *CCR2*+/+ and DMM *CCR2*-*Col1αKO* groups separately at each time point, following adjustment for multiple comparisons (Benjamini–Hochberg).

**Table 2 biomolecules-13-00891-t002:** Effect of early *Ccr2* inactivation (before DMM) on bone damage in mouse *CCR2*-*Col1αKO* and *CCR2*+/+ knee joints (N = 8).

**OA Parameters**		**Week 2**				**Week 4**			
		**Z (*p*-Value)**	**Mean Score** **(Rank Sums)**	**Median**	**Mean (SD)**	**Z (*p*-Value)**	**Mean Score** **(Rank Sums)**	**Median**	**Mean (SD)**
Osteophyte Cartilage Quantification	DMM *CCR2*+/+	−2.26 [0.08]	5.75	34.4	35.72 (13.59)	0–0.37 [0.71]	8	52.98	49.12 (21.50)
	DMM *CCR2*-*AggKO*		11.25	51.51	52.34 (13.70)		9	60.19	53.87 (23.88)
Osteophyte Size	DMM *CCR2*+/+	0.73 (0.61)	9.38	1	1.00 (0.19)	0.43 (0.66)	9.06	0.88	0.94 (0.26)
	DMM *CCR2*-*AggKO*		7.63	1	0.91 (0.33)		7.94	0.88	0.88 (0.38)
Osteophyte Maturity	DMM *CCR2*+/+	0.91 (0.36)	9.63	1	1.03 (0.45)	0.65 (0.51)	9.31	1	1.06 (0.50)
	DMM *CCR2*-*AggKO*		7.38	0.88	0.81 (0.46)		7.69	0.88	0.84 (0.33)
%BV over TV	DMM *CCR2*+/+	1.42 (0.21)	10.25	85.76	84.76 (3.03)	0.79 (0.43)	9.5	88.3	88.16 (5.38)
	DMM *CCR2*-*AggKO*		6.75	80.64	80.59 (6.61)		7.5	86.74	85.54 (3.81)
Subchondral Bone Thickness	DMM *CCR2*+/+	0–0.26 [0.87]	8.13	94,856	103,601 (21,658)	−0.16 [0.87]	8.25	130,217	136,746 (36,572)
	DMM *CCR2*-*AggKO*		8.88	105,288	119,710 (47,463)		8.75	127,770	138,730 (32,706)
**OA Parameters**		**Week 8**				**Week 12**			
		**Z (*p*-Value)**	**Mean Score (Rank Sums)**	**Median**	**Mean (SD)**	**Z (*p*-Value)**	**Mean Score (Rank Sums)**	**Median**	**Mean (SD)**
Osteophyte Cartilage Quantification	DMM *CCR2*+/+	0.47 (0.71)	9.13	42.42	43.20 (10.90)	−0.68 [0.71]	7.63	32.13	31.86 (15.99)
	DMM *CCR2*-*AggKO*		7.88	34.9	39.61 (13.39)		9.38	38.17	35.77 (11.86)
Osteophyte Size	DMM *CCR2*+/+	1.66 (0.20)	10.5	1.63	1.69 (0.62)	2.43 (0.08)	11.38	1.63	1.78 (0.41)
	DMM *CCR2*-*AggKO*		6.5	1.25	1.22 (0.25)		5.63	1.25	1.22 (0.36)
Osteophyte Maturity	DMM *CCR2*+/+	1.07 (0.28)	9.81	1.38	1.53 (0.82)	1.81 (0.07)	10.69	1.63	1.69 (0.32)
	DMM *CCR2*-*AggKO*		7.19	1	1.06 (0.32)		6.31	1.13	1.28 (0.43)
%BV over TV	DMM *CCR2*+/+	1.63 (0.20)	10.5	92.54	92.27 (4.24)	3.20 (0.004)	12.38	97.87	97.06 (3.08)
	DMM *CCR2*-*AggKO*		6.5	88.39	87.67 (5.25)		4.63	84.99	84.81 (5.55)
Subchondral Bone Thickness	DMM *CCR2*+/+	−1 (0.64)	7.25	132,176	138,215 (26,144)	3.10 (0.008)	12.25	205,757	205,979 (36,612)
	DMM *CCR2*-*AggKO*		9.75	159,877	154,085 (28,876)		4.75	113,190	118,064 (19,106)

Data presented show the median, mean and standard deviation (SD) for each group at separate time points (two, four, eight, and twelve weeks post-surgery). Indicated *z*- and *p*-values were determined by Wilcoxon rank sum tests between DMM *CCR2*+/+ and DMM *CCR2*-*Col1αKO* groups separately at each time point, following adjustment for multiple comparisons (Benjamini–Hochberg).

**Table 3 biomolecules-13-00891-t003:** Effect of late *Ccr2* inactivation (four weeks post-DMM) on cartilage damage in mouse *CCR2*-*Col1αKO* and *CCR2*+/+ knee joint (N = 8).

OA Parameters		Week 8				Week 12			
		Z (*p*-Value)	Mean Score (Rank Sums)	Median	Mean (SD)	Z (*p*-Value)	Mean Score (Rank Sums)	Median	Mean (SD)
ACS	*CCR2*+/+	−0.21 [0.83]	8.19	5.75	6.56 (2.58)	2.79 (0.01)	11.88	9.38	9.66 (1.63)
	*CCR2*-*AggKO*		8.81	6.38	6.47 (1.28)		5.13	5.5	5.53 (2.51)
Saf-O	*CCR2*+/+	0 [1.00]	8.5	7.5	7.56 (3.32)	2.88 [0.01]	10.79	10	9.75 (1.22)
	*CCR2*-*AggKO*		8.5	7.5	7.44 (0.75)		4.21	7	6.36 (1.63)
Cartilage Histomorphometry	*CCR2*+/+	−0.16 (0.87)	8.25	21,074	23,807 (10,260)	−1.11 (0.54)	7.13	20,086	19,973 (3049)
	*CCR2*-*AggKO*		8.75	26,515	24,075 (8390)		9.88	24,930	25,629 (7527)

Data presented show median, mean and standard deviation (SD) for each group at separate time points (eight and twelve weeks post-surgery). Indicated *z*- and *p*-values were determined by Wilcoxon rank sum tests between DMM *CCR2*+/+ and DMM *CCR2*-*Col1αKO* groups separately at each time point, following adjustment for multiple comparisons (Benjamini–Hochberg).

**Table 4 biomolecules-13-00891-t004:** Effect of late *Ccr2* inactivation (four weeks post-DMM) on bone damage in mouse *CCR2*-*Col1αKO* and *CCR2*+/+ knee joint (N = 8).

OA Parameters		Week 8				Week 12			
		Z (*p*-Value)	Mean Score (Rank Sums)	Median	Mean (SD)	Z (*p*-Value)	Mean Score (Rank Sums)	Median	Mean (SD)
Osteophyte Cartilage Quantification	*CCR2*+/+	0.58 [0.56]	9.25	41.93	41.62 (12.42)	−2.99 [0.006]	4.88	31.98	33.05 (6.13)
	*CCR2*-*AggKO*		7.75	38.43	38.27 (10.71)		12.13	45.78	46.90 (7.85)
Osteophyte Size	*CCR2*+/+	−0.91 [0.73]	7.38	0.75	0.88 (0.57)	−0.16 [0.87]	8.25	1	1.03 (0.36)
	*CCR2*-*AggKO*		9.63	1.13	1.13 (0.33)		8.75	1	1.06 (0.35)
Osteophyte Maturity	*CCR2*+/+	−0.06 [0.96]	8.38	0.88	0.94 (0.22)	2.55 [0.02]	11.44	1.25	1.28 (0.21)
	*CCR2*-*AggKO*		8.63	1	0.94 (0.26)		5.56	1	0.94 (0.22)
%BV over TV	*CCR2*+/+	1.73 [0.08]	10.63	91.03	89.93 (4.61)	2.26 [0.05]	11.25	96.1	94.34 (4.73)
	*CCR2*-*AggKO*		6.38	85.61	85.39 (4.08)		5.75	88.37	88.4 (2.96)
Subchondral Bone Thickness	*CCR2*+/+	1.73 [0.08]	10.63	199,278	210,692 (40,762)	2.26 [0.05]	11.25	23,0354	21,8213 (47,127)
	*CCR2*-*AggKO*		6.38	168,541	173,802 (37,031)		5.75	161,005	152,247 (40,161)

Data presented show the median, mean, and standard deviation (SD) for each group at separate time points (eight and twelve weeks post-surgery). Indicated *z*- and *p*-values were determined by Wilcoxon rank sum tests between DMM *CCR2*+/+ and DMM *CCR2*-*Col1αKO* groups separately at each time point, following adjustment for multiple comparisons (Benjamini–Hochberg).

**Table 5 biomolecules-13-00891-t005:** Effect of early *Ccr2* inactivation (before DMM) on pain assessment in mouse *CCR2*-*Col1αKO* and *CCR2*+/+ (N = 8).

Pain	Incapacitance Meter	Von-Frey Filaments		
Time PO/Stage	DMM *CCR2*+/+ *vs.* DMM *CCR2*-*Col1KO*	DMM *CCR2*+/+ vs. DMM *CCR2*-*Col1KO*	*DMM CCR2*+/+ vs. *Contralateral CCR2*+/+	*DMM CCR2*-*Col1KO* vs. *Contralateral CCR2*-*Col1KO*
4 wks/mild	0.01 (−0.04, 0.06) [*p* = 0.7480]	−0.33 (−1.03,0.37) [*p* = 0.3613]	0.04 (−0.56,0.64) [*p* = 0.9132]	0.23 (−0.45,0.91) [*p* = 0.5194]
8 wks/moderate	0.14 (0.03, 0.26) [*p* = **0.0177**]	−0.19 (−0.88,0.51) [*p* = 0.6109]	0.02 (−0.59,0.64) [*p* = 0.9455]	0.21 (−0.46,0.88) [*p* = 0.5523]
12 wks/severe	0.06 (−0.01, 0.14) [*p* = 0.0974]	−0.46 (−0.95,0.03) [*p* = 0.0627]	−0.66 (−1.14, −0.19) [*p =* **0.0064**]	0 (−0.45,0.45) [*p* = 0.9912]
16 wks/severe	0.15 (0.04, 0.26) [*p* = **0.0097**]	−0.02 (−0.4,0.36) [*p* = 0.9301]	−0.44 (−0.86,−0.02) [*p* = **0.0376**]	−0.49 (−0.91, −0.07) [*p* = **0.0230**]
20 wks/severe	0.16 (0.07, 0.24) [*p* = **0.0006**]	−0.16 (−0.54,0.22) [*p* = 0.4156]	−0.29 (−0.65,0.08) [*p* = 0.1245]	0 (−0.33,0.33) [*p* = 0.9872]

Data presented are least-squares mean differences (and corresponding 95% Cis) from two separate linear mixed effect models. The *p*-values are also reported in [], with significant values in bold.

## Data Availability

The datasets generated during the current study are available from the corresponding author upon reasonable request.

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
