# Peer review of "CC-Chemokine Receptor-2 Expression in Osteoblasts Contributes to Cartilage and Bone Damage during Post-Traumatic Osteoarthritis"

_biomolecules, 2023, doi:10.3390/biom13060891_

Round 1
Reviewer 1 Report
The aims, working hypothesis and research designs are clearly written. The conclusion is supported by the data provided except for insufficient evidence for showing the specific Cre-recombination and deletion of CCR2. Deletion of CCR2 in osteoblasts and osteoblast-linage cells reduced subchondral bone thickness is worthy to report. The data presentation and discussion can be improved as below. In addition, some minor comments are raised to improve the scientific rigor of this manuscript.
Specific comments:
Major points
1. The immunostaining of GFP in Figure 1 has non-specific staining, which disturb to support the conditional activation of Cre recombination. In addition, Supplement Figure S1 showed negative staining for the GFP proteins in subchondral bone. Is this due to low activity of Col1 promoter in osteocytes? The authors need to provide clearer results that demonstrate specific GFP staining in osteoblasts in the whole joints. Fluorescence imaging of cryosections should provide more rigorous results.
2. The data in table 1 and 2 include overlapped values with Figures 2-5. These overlapped results should be provided as supplement materials.
3. The analysis of subchondral bone is very important for this study. The microCT analysis would be better for future studies. The regions analyzed for the subchondral bone need to be specified.
4. Repeating results in Discussion should be removed. Selective effects of CCR2 deficiency on very early phase and severe phase, and on static pain, not on evoked pain are interesting. The underlying possible mechanisms for these two points should be discussed.
Minor points
1. The scores for articular cartilage pathology are provided as an average of 4 anatomical regions. The medial regions should be more affected by DMM. Therefore, the values should be provided separately for four different sites.
2. The higher magnification of the histology at the early phases should be provided, and the difference between control and CKO groups should be described. The provided histology is difficult to recognize the difference.
3. It would be nicer to provide CCR2 expression in osteoblasts underneath the articular cartilage at different stages in the DMM control group.
4. l.248, CCR2flx/flx-eGFP/AggCreERT2- should read CCR2flx-eGFP/Col1a1CreER
5. Provide the sources for the transgenic mouse strains.
Author Response
The aims, working hypothesis and research designs are clearly written. The conclusion is supported by the data provided except for insufficient evidence for showing the specific Cre-recombination and deletion of CCR2. Deletion of CCR2 in osteoblasts and osteoblast-linage cells reduced subchondral bone thickness is worthy to report. The data presentation and discussion can be improved as below. In addition, some minor comments are raised to improve the scientific rigor of this manuscript.
Specific comments:
Major points
- The immunostaining of GFP in Figure 1 has non-specific staining, which disturb to support the conditional activation of Cre recombination. In addition, Supplement Figure S1 showed negative staining for the GFP proteins in subchondral bone. Is this due to low activity of Col1 promoter in osteocytes? The authors need to provide clearer results that demonstrate specific GFP staining in osteoblasts in the whole joints. Fluorescence imaging of cryosections should provide more rigorous results.
Author’s response: we have provided a better picture for the GFP immunostaining in Figure 1. Depending on the antibody (in our case the GFP), some non-specific background might be difficult to avoid, but the positive staining is clearly detected in the nuclei of the cells.
For the Supplemental Fig S1, the Reviewer is right, osteoblasts are predominantly located in the periosteum, endosteum, and the surfaces of trabecular bone, and we have included images of that regions as well in Figure 1. In addition to that, we would like to point out that the experimental conditions of the IHC have been optimized for the specific tissue analyzed, which sometimes might be different for each antibody; the negative GFP staining in the subchondral bone might be due partially to less osteoblast activity in that region and partially to the different experimental conditions (Ab dilutions, developing time) due to the different penetration in cartilage vs bone. All the pictures compared in the same figures have been stained for the same reaction time to minimize the experimental variability. We have better clarified the methods to reflect these conditions:
Methods, page 4, line 189
- The data in table 1 and 2 include overlapped values with Figures 2-5. These overlapped results should be provided as supplement materials.
Author’s response: We noticed some missing part in Table 1 in the original version submitted as final file (corresponding to 2wk and 4wk time point), and we apologize for that. To improve the reading of the manuscript and to avoid a big single Table, we have split both Table 1 and 2 and included smaller Tables with statistical values in the paragraph of the corresponding figures. As consequence, original Table 1 is now split in new Table 1 (cartilage changes, early recombination), new Table 2 (bone changes, early recombination) and new Supplemental Table S2 (synovial hyperplasia, early recombination); original Table 2 is now split in new Table 3 (cartilage changes, late recombination), new Table 4 bone changes, late recombination) and new Supplemental Table S4 (synovial hyperplasia, late recombination).
- The analysis of subchondral bone is very important for this study. The microCT analysis would be better for future studies. The regions analyzed for the subchondral bone need to be specified.
Author’s response: we acknowledge that µCT techniques are a potent tool to evaluate bone parameters and we will consider its use in future different studies. µCT analyses require a different handling of the tissue and are not compatibles with the whole joint histology evaluation needed for osteoarthritis pathologic features, and they would require a complete separate experimental set. To minimize animal numbers we have taken advantage from a recent report (Nagira K, Ikuta Y, Shinohara M, Sanada Y, Omoto T, Kanaya H, Nakasa T, Ishikawa M, Adachi N, Miyaki S et al.: Histological scoring system for subchondral bone changes in murine models of joint aging and osteoarthritis. Sci Rep 2020, 10(1):10077) where Nagira et al established a reliable histological score system specific for subchondral bone changes in murine model of osteoarthritis. They compared results with µCT values and conclude that the system is applicable for OA changes in murine models. This comment has been raised also by Rev#2 and, as suggested, we have added a sentence in the Manuscript to acknowledge this limitation.
Discussion, page 20, line 569.
In Figures 3F and 5F we have circled the subchondral bone area calculated numerically in the graphs of Figures 3E and 5E, respectively. The area in the graphs is the average of 8 different samples. Figure legends have been updated accordingly.
- Repeating results in Discussion should be removed. Selective effects of CCR2 deficiency on very early phase and severe phase, and on static pain, not on evoked pain are interesting. The underlying possible mechanisms for these two points should be discussed.
Author’s response: we removed repeating results in the Discussion and elaborated the pain results in more details.
Page 21, line 603
Minor points
- The scores for articular cartilage pathology are provided as an average of 4 anatomical regions. The medial regions should be more affected by DMM. Therefore, the values should be provided separately for four different sites.
Author’s response: we thank the reviewer for this suggestion and have substituted the Articular Cartilage graphs using only semiquantitative scores from the medial side of the knee (femurs and tibia). Although there are not changes in the final outcomes, the data look cleaner and they better fit with the rest of the figures that were already generated using score from the medial side, which is more affected by the surgery.
Methods, Figure 2 and 4, Figure legends as well as Table 1 and 2 reflect these changes.
- The higher magnification of the histology at the early phases should be provided, and the difference between control and CKO groups should be described. The provided histology is difficult to recognize the difference.
Author’s response: we have provided magnifications of the histology in Figures 2 and 4, and added a description of changes in the figure legends. We have also added a brief description of the score in the corresponding Results.
- It would be nicer to provide CCR2 expression in osteoblasts underneath the articular cartilage at different stages in the DMM control group.
Author’s response: We have previously published the expression of CCR2 in bone during different OA stages in the DMM model in control animals, in the same strain as reported in this Manuscript (Longobardi L, Temple JD, Tagliafierro L, Willcockson H, Esposito A, D’Onofrio N, Stein E, Li T, Myers TJ, Ozkan H et al.: Role of the C-C chemokine receptor-2 in a murine model of injury-induced osteoarthritis. Osteoarthritis Cartilage 2017, 25(6):914-925.). The image is below and the reference is included in the Manuscript.
- 248, CCR2flx/flx-eGFP/AggCreERT2- should read CCR2flx-eGFP/Col1a1CreER
Author’s response: thank you for catching this mistake, it has been corrected.
- Provide the sources for the transgenic mouse strains.
Author’s response: The source is listed in the Acknowledgments and has been added in the Supplemental methods, Appendix A.

Reviewer 2 Report
Biomolecules 2316240
In this manuscript, the authors characterized the PTOA phenotypes after inducing CCR2 depletion in the bone tissues in the murine model. Specifically, the authors induced CCR2 depletion before (early) and after (late) DMM surgery and demonstrated that CCR2 depletion in the bone tissues protects against cartilage damage and reduces subchondral bone sclerosis at severe stages of PTOA. In addition, the authors demonstrated that CCR2 depletion in the bone tissue does not affect synovitis and induced pain in the joint.
However, there are several flaws within this manuscript
Major comments:
1. Line 84-85, the authors stated that the purpose of their study is to determine the temporal role of CCR2 in bone during PTOA. While the authors assessed PTOA phenotypes 2wk, 4wk, 8wk, and 12wk post DMM surgery, all the statistical comparison was done between the CCR2 knockout and control groups. The authors should include the comparisons between different timepoints within the same genotype.
2. It is surprising that PTOA phenotypes are similar regardless of depleting CCR2 before or after DMM surgery. In addition, the ACS score and Safranin-O score are comparable between early and late CCR2 depletion. While the authors briefly speculated the underlying mechanisms in Line 484-486, including statistical comparisons between the early and late CCR2 depletion and elaborating on the potential mechanisms would improve this manuscript.
3. The immunostaining in Figure 1 showed that the CCR2 is depleted within the cortical bone, while osteoblasts are predominantly located in the periosteum, endosteum, and the surfaces of trabecular bone. The authors should include images of these areas.
4. In Figure2, the authors presented the H&E and Saf-O staining of the knee joint after DMM surgery. It seems that cartilage loss is more severe 8wk post-DMM compared to 12wk in the CCR2-Col1aKO group, which is also reflected in the ACS score and Saf-O score. However, the authors did not perform statistical analysis of the ACS score and Saf-O score longitudinally within the same genotype. And it is not clear whether depletion of CCR2 in the bone tissue would promote cartilage regeneration.
Minor comments
1. In the upper right panel of Figure 1B, the positive IHC staining for GFP indicated by the red arrows is not very clear. The authors should provide better contrasted images.
2. The subchondral bone analysis was done using 2D measurements, which is highly dependent on the section that was collected, instead of 3D measurements such as micrIn this manuscript, the authors characterized the PTOA phenotypes after inducing CCR2 depletion in the bone tissues in the murine model. Specifically, the authors induced CCR2 depletion before (early) and after (late) DMM surgery and demonstrated that CCR2 depletion in the bone tissues protects against cartilage damage and reduces subchondral bone sclerosis at severe stages of PTOA. In addition, the authors demonstrated that CCR2 depletion in the bone tissue does not affect synovitis and induced pain in the joint.
However, there are several flaws within this manuscript
Major comments:
1. PTOA phenotype has been explored in Ccr2 null mice previously and has shown varying protective effects. While in this manuscript, the authors explored bone-specific roles of CCR2 in PTOA, including references to previous research is critical.
2. Line 84-85, the authors stated that the purpose of their study is to determine the temporal role of CCR2 in bone during PTOA. While the authors assessed PTOA phenotypes 2wk, 4wk, 8wk, and 12wk post DMM surgery, all the statistical comparison was done between the CCR2 knockout and control groups. The authors should include the comparisons between different timepoints within the same genotype.
3. It is surprising that PTOA phenotypes are similar regardless of depleting CCR2 before or after DMM surgery. In addition, the ACS score and Safranin-O score are comparable between early and late CCR2 depletion. While the authors briefly speculated the underlying mechanisms in Line 484-486, including statistical comparisons between the early and late CCR2 depletion and elaborating on the potential mechanisms would improve this manuscript.
4. The immunostaining in Figure 1 showed that the CCR2 is depleted within the cortical bone, while osteoblasts are predominantly located in the periosteum, endosteum, and the surfaces of trabecular bone. The authors should include images of these areas.
5. In Figure2, the authors presented the H&E and Saf-O staining of the knee joint after DMM surgery. It seems that cartilage loss is more severe 8wk post-DMM compared to 12wk in the CCR2-Col1aKO group, which is also reflected in the ACS score and Saf-O score. However, the authors did not perform statistical analysis of the ACS score and Saf-O score longitudinally within the same genotype. And it is not clear whether depletion of CCR2 in the bone tissue would promote cartilage regeneration.
6. While it is interesting that depleting CCR2 before or after DMM surgery has protective effects on severe stage of PTOA, it is unclear how this finding would translate to a clinical setting, especially with CCR2 depletion in chondrocytes is more effective before DMM surgery at early stage OA (Line478-479). I suggest the authors including a paragraph to discuss this issue.
Minor comments
1. In the upper right panel of Figure 1B, the positive IHC staining for GFP indicated by the red arrows is not very clear. The authors should provide better contrasted images.
2. The subchondral bone analysis was done using 2D measurements, which is highly dependent on the section that was collected, instead of 3D measurements such as microCT. The authors should point out the limitation of their methods.oCT. The authors should point out the limitation of their methods.
Author Response
Reviewer #2:
In this manuscript, the authors characterized the PTOA phenotypes after inducing CCR2 depletion in the bone tissues in the murine model. Specifically, the authors induced CCR2 depletion before (early) and after (late) DMM surgery and demonstrated that CCR2 depletion in the bone tissues protects against cartilage damage and reduces subchondral bone sclerosis at severe stages of PTOA. In addition, the authors demonstrated that CCR2 depletion in the bone tissue does not affect synovitis and induced pain in the joint.
Specific comments:
Major comments:
- PTOA phenotype has been explored in Ccr2 null mice previously and has shown varying protective effects. While in this manuscript, the authors explored bone-specific roles of CCR2 in PTOA, including references to previous research is critical.
Author’s response: CCR2 is expressed in several tissues, such as monocytes, chondrocytes, osteoblasts, DRG and several other tissues. With respect to OA, studies using the DMM model in global Ccr2-null mice demonstrated a key role for CCR2 in establishing OA pain but observed a modest reduction in chondropathy, evident only at the most severe stages; to our knowledge, the specific role of Ccr2 in bone damage during PTOA has not been investigated by other researcher. As we previously noted, there are limitations in using Ccr2 global germ-line deletion as a model system for OA, as these mice develop pre- and post-natal growth plate defects. We also previously reported that CCR2 blockade caused impaired limb ossification during embryogenesis and found evidence post-natally of disorganized proliferative columns in the growth plate of Ccr2-null mice, published in a previous manuscript (Willcockson H, Ozkan H, Arbeeva L, Mucahit E, Musawwir L, Longobardi L: Early ablation of Ccr2 in aggrecan-expressing cells following knee injury ameliorates joint damage and pain during post-traumatic osteoarthritis. Osteoarthritis Cartilage 2022, 30(12):1616-1630). Therefore, altered bone remodeling may mask amelioration of OA, especially when referring to bone damage. This is already included in the Discussion. To our knowledge, our study is the first report exploring a specific role for Ccr2 in osteoblast in the whole joint pathogenesis and pain.
Discussion, page 19, line 502
- Line 84-85, the authors stated that the purpose of their study is to determine the temporal role of CCR2 in bone during PTOA. While the authors assessed PTOA phenotypes 2wk, 4wk, 8wk, and 12wk post DMM surgery, all the statistical comparison was done between the CCR2 knockout and control groups. The authors should include the comparisons between different time points within the same genotype.
Author’s response: we have reformulated the sentence at line 84-85 and eliminated the word “temporal”, as it might generate confusion. The scope of the work is to analyze the role of Ccr2 in osteoblasts during PTOA and investigate whether Ccr2 ablation delays disease onset (achieved by early ablation before injury occurs) or slows disease progression (late ablation when OA is already started), compared to controls that have an intact copy of Ccr2. We report PTOA outcomes at different stages to understand what stage is more affected, compared to controls with intact Ccr2. Ablation of PTOA can determine a less severe disease phenotype, or a slower progression, compared to Control. As suggested by the reviewer, in the new version we have included a new analysis within each genotype for the PTOA outcomes that are affected by the Ccr2 ablation (ACS, Histomorphometric analyses of cartilage quantification, Saf-O, Subchondral Bone thickness and BV/TV), for both the early (Supplemental Figure S2 and Supplemental Tale S1) and Late recombination (Supplemental Figure S4 and Supplemental Tale S3), to highlight the OA progression pattern for the affected outcomes.
We refer to these figures and Tables in the Results:
Page 9, line 294
Page 10, line 323
Page 13, line 372
Page 15, line 412
- It is surprising that PTOA phenotypes are similar regardless of depleting CCR2 before or after DMM surgery. In addition, the ACS score and Safranin-O score are comparable between early and late CCR2 depletion. While the authors briefly speculated the underlying mechanisms in Line 484-486, including statistical comparisons between the early and late CCR2 depletion and elaborating on the potential mechanisms would improve this manuscript.
Author’s response: as suggested by the reviewer, we have included a comparison between early and late inactivation for each parameter (Supplemental Fig. S6) and noticed that the phenotypes relating to both inactivation are indeed very similar, and there is no significant changes in any of the parameters at any of the time points (Supplemental Table S7); this result is not particularly surprising, as when we analyze cartilage and bone changes following early Ccr2 inactivation, all parameters are significantly improved mostly at the late stage (12weeks), with only some trend noticeable at earlier stages in some of the outcomes. This is also visualized when data are plotted by genotypes in Supplemental figures S2 and S4.
We have added a comment in the Discussion as suggested (we believe the reviewer was referring to original Line 584-586).
Discussion, Page 20, line 549
- The immunostaining in Figure 1 showed that the CCR2 is depleted within the cortical bone, while osteoblasts are predominantly located in the periosteum, endosteum, and the surfaces of trabecular bone. The authors should include images of these areas.
Author’s response: as suggested, we included new images in Figure 1.
- In Figure2, the authors presented the H&E and Saf-O staining of the knee joint after DMM surgery. It seems that cartilage loss is more severe 8wk post-DMM compared to 12wk in the CCR2-Col1aKO group, which is also reflected in the ACS score and Saf-O score. However, the authors did not perform statistical analysis of the ACS score and Saf-O score longitudinally within the same genotype. And it is not clear whether depletion of CCR2 in the bone tissue would promote cartilage regeneration.
Author’s response: Figure 2 A and B show representative sections of the surgical model in mice that are analyzed cross-sectional, since it is not possible to obtain histological longitudinal sections in the same animals. Therefore, phenotypes at 8 weeks and 12 weeks can sometime be very similar, it depends on the experimental variability of the surgery combined with animal susceptibility to OA. Longitudinal analyses cannot be provided with this method. However, we have provided other samples for 12wks time point to better reflect the numerical data. In addition, in Figure 2, we have produced enlargements of the images and highlighted the OA features. As we stated in the manuscript, ablation of CCR2 can only translate in reducing or slowing the disease progression, but cartilage does not have the capability to regenerate itself.
As stated in Comment 2, we have included cross-sectional analysis within genotypes.
- While it is interesting that depleting CCR2 before or after DMM surgery has protective effects on severe stage of PTOA, it is unclear how this finding would translate to a clinical setting, especially with CCR2 depletion in chondrocytes is more effective before DMM surgery at early stage OA (Line478-479). I suggest the authors including a paragraph to discuss this issue.
Author’s response: as suggested we have better described the importance of Ccc2 targeting as OA therapy in a potential clinical setting.
Discussion, Page 21, lines 619.
Minor comments
- In the upper right panel of Figure 1B, the positive IHC staining for GFP indicated by the red arrows is not very clear. The authors should provide better contrasted images.
Author’s response: we have included better images in Figure 1.
- The subchondral bone analysis was done using 2D measurements, which is highly dependent on the section that was collected, instead of 3D measurements such as microCT. The authors should point out the limitation of their methods.oCT. The authors should point out the limitation of their methods.
Author’s response: we acknowledge that µCT techniques are a potent tool to evaluate bone parameters and we will consider its use in future different studies. µCT analyses require a different handling of the tissue and are not compatibles with the whole joint histology evaluation needed for osteoarthritis pathologic features, and they would require a complete separate experimental set. To minimize animal numbers we have taken advantage from a recent report (Nagira K, Ikuta Y, Shinohara M, Sanada Y, Omoto T, Kanaya H, Nakasa T, Ishikawa M, Adachi N, Miyaki S et al.: Histological scoring system for subchondral bone changes in murine models of joint aging and osteoarthritis. Sci Rep 2020, 10(1):10077) where Nagira et al established a reliable histological score system specific for subchondral bone changes in murine model of osteoarthritis. They compared results with µCT valuses and conclude that the system is applicable for OA changes in murine models. This comment has been raised also by Rev#2 and, as suggested, we have added a sentence in the Manuscript to acknowledge this limitation.
Discussion, page 20, line 569.

Round 2
Reviewer 1 Report
The revised version has been improved and incorporated this reviewer's comment sufficiently. This manuscript contains enough useful and new information for researchers in the field, and may also provide new insights for the general reader.
Reviewer 2 Report
I have no further comments.